# Occupational Exposure to Ultrafine Particles and Placental Histopathological Lesions: A Retrospective Study about 130 Cases

**DOI:** 10.3390/ijerph182312719

**Published:** 2021-12-02

**Authors:** Anaïs Pasquiou, Fanny Pelluard, Guyguy Manangama, Patrick Brochard, Sabyne Audignon, Loïc Sentilhes, Fleur Delva

**Affiliations:** 1Department of Pathology, Bordeaux University Hospital, 33076 Bordeaux, France; fanny.pelluard@chu-bordeaux.fr; 2BaRITOn, INSERM U1053, University of Bordeaux, 33076 Bordeaux, France; 3ARTEMIS Center, Bordeaux University Hospital, 33076 Bordeaux, France; guyguy.manangama-duki@u-bordeaux.fr (G.M.); patrick.brochard@chu-bordeaux.fr (P.B.); sabyne.audignon@u-bordeaux.fr (S.A.); fleur.delva@chu-bordeaux.fr (F.D.); 4Bordeaux Population Health Research Center, Inserm UMR1219-EPICENE, University of Bordeaux, 33076 Bordeaux, France; 5Department of Obstetrics and Gynecology, Bordeaux University Hospital, 33076 Bordeaux, France; loic.sentilhes@chu-bordeaux.fr

**Keywords:** placenta, particulate matter, occupational exposure, nanoparticles, pathology

## Abstract

Ultrafine particles (UFPs) are particles smaller than 100 nanometers that are produced unintentionally during human activities or natural phenomena. They have a higher biological reactivity than bigger particles and can reach the placenta after maternal exposure. One study has shown an association between maternal occupational exposure to UFPs and fetal growth restriction. Yet few studies have focused on the effects of UFP exposure on placental histopathological lesions. The aim of this study was to investigate the association between maternal occupational exposure to UFPs and histopathological lesions of their placenta. The analyses were based on data from the ARTEMIS Center. A job-exposure matrix was used to assess occupational exposure to UFPs. The histopathological placental exam was performed by two pathologists who were blinded to the exposure of each subject. The examination was conducted in accordance with the recommendations of the Amsterdam consensus. The study sample included 130 placentas (30 exposed, 100 unexposed). Maternal occupational exposure to UFPs during pregnancy is significantly associated with placental hypoplasia (the phenomenon affected 61% of the exposed patients and 34% of the unexposed ones, *p* < 0.01). Further research is needed to explain its pathophysiological mechanisms.

## 1. Introduction

Particulate matter (PM) aerosols can be classified depending on their aerodynamic diameter [1]. The PM with a diameter smaller than 100 nm are usually called manufactured nanoparticles (MNPs) when they are produced intentionally (for electronics, cosmetics, etc.) and ultrafine particles (UFPs) when they are a byproduct of human activities (produced by thermic motors, arc welding, incineration, but also by electric motors, printers, etc.) or when they are produced during natural phenomena (volcano eruptions, forest fires, marine emissions, etc.) [2]. MNPs and UFPs share similar characteristics [3]. Because of their small size, their physicochemical properties differ from the bigger particles that have an otherwise similar nature [4,5]. Indeed, the proportion of surface atoms increases in relation to the atoms on the inside when the particle’s diameter decreases. This leads to an increase in the surface energy of these particles, which makes them more thermodynamically unstable. UFPs represent a small percentage of the particulate matter’s mass, but a big percentage of the number of particles [4]. After inhalation by a person, their small size allows them to travel into the pulmonary alveolae, causing local inflammation, and to cross the alveolae–capillary barrier to reach the general circulation system and distant organs. Their adverse effects on the respiratory and cardiovascular systems have been outlined in some toxicological and epidemiological studies [6,7,8]. In a recent review, several toxicological and ex vivo studies showed that UFPs can reach the placenta and the fetus [9]. Recently, a study showed the presence of ultrafine carbon particles in human placenta after the exposure of the pregnant mothers to atmospheric pollution [10]. Another study showed a positive association between maternal occupational exposure to UFPs during pregnancy and fetal growth restriction (FGR) of the child [11]. The placenta is a transitory organ, but it is essential to an uneventful pregnancy and the good development of the fetus. Placental dysfunction is associated with adverse pregnancy outcomes [12,13]. A study on the placental histological lesions of women who are exposed to UFPs could help demonstrate their role in causing some adverse obstetrical outcomes, and could facilitate the identification of some of the mechanisms of toxicity that occur during pregnancy. Exposure is easier to assess retrospectively in a professional environment than in an extra-professional environment, using existing tools, such as exposure matrices [14]. Additionally, according to a recent meta-analysis, workers’ exposure to ultrafine particles may be significantly higher than their non-occupational exposure [15]. The aim of this study was to investigate the association between the occupational exposure to UFPs of pregnant women and the histopathological lesions of their placenta.

## 2. Materials and Methods

### 2.1. ARTEMIS Center

The design of the ARTEMIS Center has been previously described by Delva, et al. [16]. Briefly, the ARTEMIS Center (“Aquitaine ReproducTion Enfance Maternité et Impact en Santé environnement”) is part of the Bordeaux University Hospital (BUH). It is dedicated to the prevention of environmental hazards affecting the reproductive functions. After a thorough review of the scientific literature and regulation reports to identify potential risk factors for reproductive functions, they prioritize the risks according to the existing degree of evidence in the literature, and link each of them to potential occupational and non-occupational exposures. So far, twelve families of risk factors have been identified, including UFP exposure. Patients are referred to the ARTEMIS Center by volunteer hospital practitioners, obstetricians, gynecologists or pediatricians based in the Nouvelle-Aquitaine region (south-west of France) if they are identified as having a condition affecting the reproduction or the course of the pregnancy. This includes fertility issues, hypertensive disorders, diabetes mellitus, premature rupture of membranes, preterm labor, placental disorder, congenital malformations, chromosomic abnormalities, fetal loss and stillbirth. The patients benefit from a medical consultation and a standardized interview conducted by a nurse, a medical resident, or an environmental health engineer. During this interview, they answer a questionnaire that includes all of the potential circumstances of exposure to the risk factors previously identified. It includes questions on the couple’s socio-demographic characteristics (age, education, home address), their job before, during and after the pregnancy (occupation, industry sector), their lifestyle habits (including alcohol and tobacco consumption, use of chemicals, cleaning and cooking habits, etc.) and their medical history. After a multidisciplinary staff meeting between the nurse, environmental health engineer, occupational physician and public health physician, targeted preventive measures for the patients are decided upon and the data that have been collected are coded in a database for future access and research. Between February 2016 and May 2018, 954 patients (624 women and 330 men) consulted at the ARTEMIS Center. Of these, 39% of them were addressed for infertility, 60% for pregnancy complications and 3% for another reason (patients could be addressed for more than one reason). 

### 2.2. Study Participants

The women who had a consultation at the ARTEMIS Center for an obstetrical pathology or a fetal malformation from May 2016 to May 2018 were included in the study if their placenta underwent a histopathological examination at the fetopathology department of Bordeaux University Hospital, France. A placental histopathological examination is performed if justified by a maternal or fetal condition, or if the placenta is macroscopically abnormal, following the recommendations made by the College of American Pathologists [17]. Maternal conditions include a range of disorders, such as suspicion of infection, pre-eclampsia, uncontrolled diabetes, etc. Fetal conditions include premature delivery, intrauterine growth restriction, signs of infection, suspicion of aneuploidy, etc. The patients whose placenta was not interpretable (e.g., if there was a problem with the conservation of the slides or with the sampling during the macroscopical exam) were excluded. The patients whose pregnancy ended with a fetal loss were also excluded, because histopathological lesions caused by the fetus maceration can be difficult to differentiate from fetal vascular malperfusion [13]. Occupational and non-occupational exposure to UFPs was then assessed. The patients were excluded if the data concerning the professional exposure were missing. 

### 2.3. Evaluation of the Non-Occupational Exposure to UFPs 

We considered exposure to UFPs contained in atmospheric pollution and in tobacco smoke to be non-occupational. Other potential exposures to domestic UFPs (such as the ones produced during cooking, heating, or cleaning) were considered to be negligible compared to professional exposure and were therefore not considered. Data concerning the patient’s address and tobacco consumption were based on the ARTEMIS Center’s database. The patient’s address allowed us to divide the location of the patient’s home into three groups: urban, semi-urban and rural, according to the “Typology of the French Countryside” [18]. This was the best way to approximate the exposure to atmospheric pollution, given the retrospective design of our study and the data available. Patients who declared having never smoked or who stopped smoking tobacco before the beginning of the pregnancy were considered to be non-smokers. The others were considered to be smokers. Passive tobacco exposure was also based on the patient’s declaration. 

### 2.4. Evaluation of the Occupational Exposure to UFPs 

To assess the occupational exposure to UFPs, we used the MatPUF job-exposure matrix (JEM), which is specific to UFPs. The methodology of its construction has been previously described by Audignon, et al. [19]. Briefly, work processes that lead to an exposure to UFPs were identified thanks to a comprehensive literature review combined with the judgement of an expert panel that covered various domains (toxicology, industrial hygiene, physics, atmospheric chemistry and epidemiology). These work processes were linked to occupations, as defined by the International Standard Classification of Occupation (ISCO), 1968 edition. Then, two experts in industrial hygiene evaluated the probability of exposure for each association between an occupation and a work process. In the case where several work processes were involved, a summarized probability was calculated. The probability was weighted depending on the industry sector, as defined by the French Nomenclature of Activities (NAF), 2000 edition. Finally, the JEM provided a probability of exposure for a given occupation, which was defined as the proportion of individuals exposed to UFPs through the implementation of work processes that were likely to lead to an exposure. It was expressed as a percentage on a semi-quantitative scale and then grouped into four categories: occupationally unexposed (0%), possible (>0–10%), probable (>10–50%) and very probable (>50%) exposure. The unexposed category implies that exposure in the considered occupation was no higher than the level of exposure faced by the general population.

We performed a retrospective exposure assessment based on the job descriptions provided by the patients during their interview at the ARTEMIS Center. Given that the occupations were coded in the ARTEMIS database according to the ISCO 2008 edition [20] and industry sectors were coded according to the NAF 2008 edition [21], we first transcoded them into the ISCO 1968 edition and the NAF 2000 edition, using a correspondence table. We then linked these jobs to the JEM, which gave us an estimation of the probability of exposure to UFPs for the job that the mother held during pregnancy. We classified the mother into two groups: exposed (job held during pregnancy associated with a probability of exposure > 0, *n* = 28) and unexposed (job held during pregnancy associated with a probability of exposure ≤ 0, *n* = 86).

### 2.5. Evaluation of the Histological Placental Lesions 

After a delivery at the BUH maternity ward, placentas that were to undergo a histopathological exam were sent to the BUH fetopathology department, where a standard procedure was performed by a fetopathologist or a technician trained in macroscopy. After at least 48 h hours of fixation in 10% neutral buffered formalin and after the removal of the cord and membranes, the placenta was measured, weighed and any macroscopical anomaly was noted. For singleton pregnancies, at least four samples were embedded in paraffin blocs, targeting the macroscopical lesions if present. The samples included two sections of the umbilical cord, a roll of the membranes and at least three slices of placental parenchyma that contained the fetal and the maternal sides of the placenta. For twin pregnancies, at least nine samples were taken, one containing a roll of the interamniotic membrane and one containing two sections of the umbilical cord and a roll of the membranes for each of the fetuses. The microscopic examination was performed by a senior pathologist (the author, F.P.) and a resident in pathology (the author, A.P.), who were both blinded to whether the mother was exposed to UFPs. The data included in the medical file of the patients were known. The histopathological lesions were classified using the classifications determined by the Amsterdam Placental Workshop Group [13]. Maternal stromal-vascular malperfusion was defined as a composite criterion that included placental hypoplasia (placental weight under the 10th centile and/or a cord diameter under 8 mm at term), villous infarction (any infarction before 34 weeks of gestation (WG) or any non-peripheral infarction involving 5% or more of the parenchyma at term), abruptio placentae, distal villous hypoplasia (the paucity of villi in the lower two-thirds of the placenta, involving 30% or more of one full-thickness parenchymal slide), accelerated villous maturation (a diffuse pattern of term-appearing villi with increased syncytial knots (present in at least 33% of the villi) and intervillous fibrin) and decidual arteriopathy (acute atherosis, fibrinoid necrosis with or without foam cells, mural hypertrophy, chronic perivasculitis, the absence of spiral artery remodeling, arterial thrombosis and/or persistence of intramural endovascular trophoblast in the third trimester). Fetal stromal-vascular malperfusion was defined as a composite criterion that included the thrombosis of an umbilical cord, chorionic plate or stem villi vessel, avascular villi (three or more foci of two or more terminal villi with a total loss of villous capillary and fibrosis of the stroma), intramural fibrin deposition (fibrin or fibrinoid deposition (subendothelial or intramuscular) within the wall of large fetal vessels, with or without calcification), villous stromal-vascular karyorrhexis (three or more foci of two or more terminal villi that showed the karyorrhexis of fetal cells (nucleated erythrocytes, leukocytes, endothelial cells and/or stromal cells) with the preservation of the surrounding trophoblast) and stem vessel obliteration (a marked thickening of the vessel wall and resultant obliteration of the vascular lumen). Delayed villous maturation was defined as a monotonous villous population (at least ten villi) with centrally placed capillaries and decreased vasculosyncytial membranes, present in at least 30% of one full-thickness parenchymal slide. Ascending intrauterine infection was defined as a composite criterion composed of maternal inflammatory response (MIR) and fetal inflammatory response (FIR). MIR was divided into three stages: acute subchorionitis, which is defined as the presence of neutrophils in the subchorial intervillous space or beneath the chorion laeve layer; acute chorioamnionitis, which is defined as the presence of neutrophils in the fibrous chorion and/or amnion; and necrotizing chorioamnionitis, which is defined as a stage two aspect with karyorrhexis of polymorphonuclear leukocytes, amniocyte necrosis, and/or amnion basement membrane hypereosinophilia. FIR was divided into three stages: chorionic vasculitis or umbilical phlebitis, the involvement of the umbilical vein and one or more umbilical arteries, and necrotizing funisitis. Defining inflammatory lesions involved a composite criterion, with the identification of a villitis of unknown etiology (lymphohistiocytic inflammation in the stroma of terminal villi, without an identified etiology, involving at least two separate foci of villi), eosinophilic/T-cells vasculitis (T lymphocytes and eosinophils infiltrating a chorionic plate vessel), chronic intervillositis (the infiltration of the intervillous space by histiocytes) and chronic deciduitis (chronic inflammation with plasma cells within the basal plate). We chose to review the two placentas from the dichorionic diamniotic pregnancies separately because the histological aspect of the two placentas from the same pregnancy can be very different. The placenta of singleton and monochorionic pregnancies were counted as one entry.

### 2.6. Population Characteristics

Data concerning the maternal age, type of pregnancy, obstetrical pathologies and term of pregnancy were collected from the medical files of the obstetrics department, based on the physician’s documentation. Data concerning the body mass index (BMI), level of maternal education, medical history, tobacco exposure during pregnancy and home address were collected from the ARTEMIS Center database. 

### 2.7. Statistical Analysis

Data were analyzed using pvalue.io, a Graphic User Interface for the R statistical analysis software for scientific medical publications (Medistica), 2019 [22]. Continuous variables were calculated as a mean +/– standard deviation. Categorical variables were calculated as a rate (%). Chi square or Fisher’s tests were used accordingly, depending on their criteria. A *p* value of <0.05 was considered to be statistically significant. Given the small number of patients in this study, we chose not to perform multivariate analyses. 

## 3. Results

### 3.1. Population

As shown in the flow chart (Figure 1), of the 169 patients who consulted at the ARTEMIS Center for an obstetrical pathology or a fetal malformation, 114 of them had an assessment of the histopathologic lesions of their placenta and of their occupational exposure to UFPs. Of these, 28 were classified as exposed to UFPs and 86 as unexposed, corresponding respectively to 30 and 100 placentas (the placenta of each twin was reviewed independently in dichorionic diamniotic pregnancies).

Characteristics of the participants are shown in Table 1 and Table 2. The mean maternal age was 31.9 years (SD: 5.62). The exposure to tobacco during pregnancy concerned approximately 27% of the patients, with 18% considered to be active smokers and approximately 9% to be exposed to passive smoking. None of the patients declared smoking cannabis. Of these patients, 70% lived in an urban area. All of the pregnancies were pathological. The three main obstetrical pathologies were preterm birth (92% of the children were born before 37 WG, and 52% before 32 WG), FGR (44%) and pre-eclampsia (32%). None of the patients suffered from non-gestational diabetes. The level of maternal education was significantly different between the exposed and the unexposed groups. Indeed, 68% of the unexposed group had a short-cycle higher education, a master’s degree or above, whereas this was the case for only 21.1% of the exposed group. An upper-level education is associated with office jobs, which are less exposed to UFPs. 

### 3.2. Exposure

The sources of the exposure to UFPs are described in Table 3. There were 28 patients who were identified as professionally exposed (i.e., the probability of exposure was greater than zero). The most frequent occupation held in the exposed group was “cleaner” (*n* = 10), followed by childcare professional (*n* = 5), agricultural worker (*n* = 3) and cook (*n* = 2).

### 3.3. Histological Lesions

Placental lesions are described in Table 4 and Table 5. There was a significant association between maternal occupational exposure to UFPs and placental hypoplasia (34% in the unexposed group vs 61% in the exposed group, *p* < 0.01, Chi-squared). There was no significant difference between the non-exposed group and the exposed group concerning the four histological syndromes: maternal stromal-vascular malperfusion lesions (90% and 100%, respectively; *p* = 0.23), fetal stromal-vascular malperfusion lesions (26% and 13%, respectively; *p* = 0.12), ascending intra-uterine lesions (20% and 6.5%, respectively; *p* = 0.08) or immune lesions (20% and 6.5%, respectively; *p* = 0.08). 

## 4. Discussion

This study suggests that maternal occupational exposure to UFPs during pregnancy is associated with a higher risk of placental hypoplasia compared to unexposed patients (61% and 34%, respectively; *p* < 0.01). There was no association between the mother’s occupational exposure to UFPs and placental histological lesions (maternal vascular malperfusion of the placental bed, fetal vascular malperfusion of the placental bed or immune/infectious lesion). 

Placental hypoplasia is described as the umbilical cord having a low placental weight and/or diameter compared to where it should be according to the term of the pregnancy. Low placental weight and a thin umbilical cord are associated with FGR [23,24,25]. This finding is consistent with the study conducted by Manangama, et al. [11], which showed an association between occupational exposure to UFPs and FGR, using the same JEM as in this study. Several toxicological studies show the effect of ultrafine particles on the placenta. A recent meta-analysis by Bongaerts, et al. [9] showed that many kinds of nanoparticles can reach the placenta, accumulate in it and/or cross it to reach the fetus, including airborne particles. Other studies suggest an association between maternal exposure to nanoparticles and histological lesions of the placenta, including an alteration of the placental weight, a decrease in the placental vascularization and an increase in the amount of fibrin deposits [26,27,28]. There are, however, no studies that show an isolated decrease in the placental weight without being accompanied by histological lesions. Several hypotheses can explain these results. First, several studies suggest that nanoscale particles can potentially induce oxidative stress and epigenetic alterations due to their physicochemical properties [29,30]. Saenen, et al. suggest that the mother’s exposure to air pollution and PM2.5 (particulate matter <2.5 µm, which includes UFPs) is associated with some variations in the placental epigenetic mechanisms, such as global DNA methylation or the levels of expression of miRNA [31]. Other studies suggest that some alterations in the expression of miRNA could be associated with higher risks of pre-eclampsia [32] or FGR [33]. These mechanisms, which happen on the molecular scale, could explain why no specific placental histological lesion was statistically significant in our study. Furthermore, the classification of the histopathological lesions that we used only considers lesions which are quite pronounced. It is possible that exposure to UFPs is associated with histological changes that are not severe enough to be classified as a histological lesion in this classification. This would lead to an underestimation of the association between the exposure to UFPs and the presence of histological placental lesions. Our results could also be explained by methodological issues. In this study, all the pregnancies were pathological. Indeed, if the pregnancy is physiological, the placenta doesn’t undergo a histopathological examination, so the patient can’t be included in this study. It is possible that the lesions associated with exposure to UFPs are blended with the lesions associated with other pathologies and, therefore, are not revealed to be statistically different in the context of these two groups. Furthermore, to classify the patients into the exposed group, we chose the cut-off as the “probability of exposure >0%”. It is possible that among this group, some patients are not exposed, which decreases the probability of showing a difference between the two groups. It is worth noting that the association between UFP exposure and placental hypoplasia would be even more apparent if these factors were absent.

The present study has several strengths. The placenta samples underwent a double review by the same two trained pathologists, who were blinded to the UFP exposure status of the patients. We used the Amsterdam Placental Workshop Group criteria, which is the most recent consensus concerning the placental histopathological examination, and these criteria are used daily in pathological examinations. We performed a standardized exposure assessment using the MatPUF JEM. In this retrospective study, it prevents recollection bias by the patients of their potential professional exposure.

This study has several limitations. We did not perform multivariable analysis because of the limited sample size. Therefore, there could be some confusion bias that we did not consider in the statistical analysis. However, the main risk factors that lead to a small placenta are hypertensive disorders before or during the pregnancy, including pre-eclampsia, severe malnutrition, tobacco smoking during pregnancy, a low weight intake during the pregnancy and pre-existing diabetes with vascular lesions [23]. There was no significant difference between the two groups concerning the hypertensive disorders, tobacco smoking habits, diabetes, and BMI of the patients. We did not have data about the weight intake during pregnancy.

The use of the JEM MatPUF to assess the exposure to UFPs can lead to three methodological biases. First, as it is a job-exposure matrix, it does not consider the potential extra-professional exposure to UFPs. There was no significant difference between the two groups concerning tobacco smoking or residential area, which are potential cofounders for UFP exposure. However, the patients could be exposed to other sources of UFPs, and particularly at home during pregnancy (e.g., during cooking and cleaning). This could lead to the presence of confounding bias. Second, as with all exposure matrices, it only considers homogenous groups of exposure based on the occupation that the patient had during the pregnancy. It does not consider their exposure on an individual level, which could lead to a differential misclassification and an underestimation of the association [34]. Third, the design of the current version of MatPUF [19] does not provide a metric for the level of intensity of exposure. Therefore, it was not possible to examine the effects associated with high- or low-intensity exposure. Moreover, the small size of the exposed group did not allow us to refine the analysis with other data, such as the length and period of the occupational exposure to UFPs (e.g., during the first, second or third trimester of pregnancy). Some studies suggest greater adverse effects on fetal growth and low birth weight after an exposure of the mother during the second and third trimester of the pregnancy, either to particulate matter [35,36] or to carbonaceous particles [37,38]. These results might be explained by a saturation of the placental mechanisms of compensation after a certain length of exposure to UFPs. It might also be explained by a longer professional exposure to the particulate matter. It would be interesting to consider the time and length of exposure in further studies.

## 5. Conclusions

There is a growing concern regarding the mother’s exposure to ultrafine particles during pregnancy and adverse obstetrical outcomes. Yet there is a lack of data in human populations on the effects of this exposure on the morbidity of the fetus and the mother. In this study, we showed a significant association between maternal occupational exposure to UFPs and placental hypoplasia. These results are consistent with existing epidemiological and toxicological studies. This should encourage further studies to examine the effects of UFPs on the placenta, both on the histological and on the molecular level.

## Figures and Tables

**Figure 1 ijerph-18-12719-f001:**
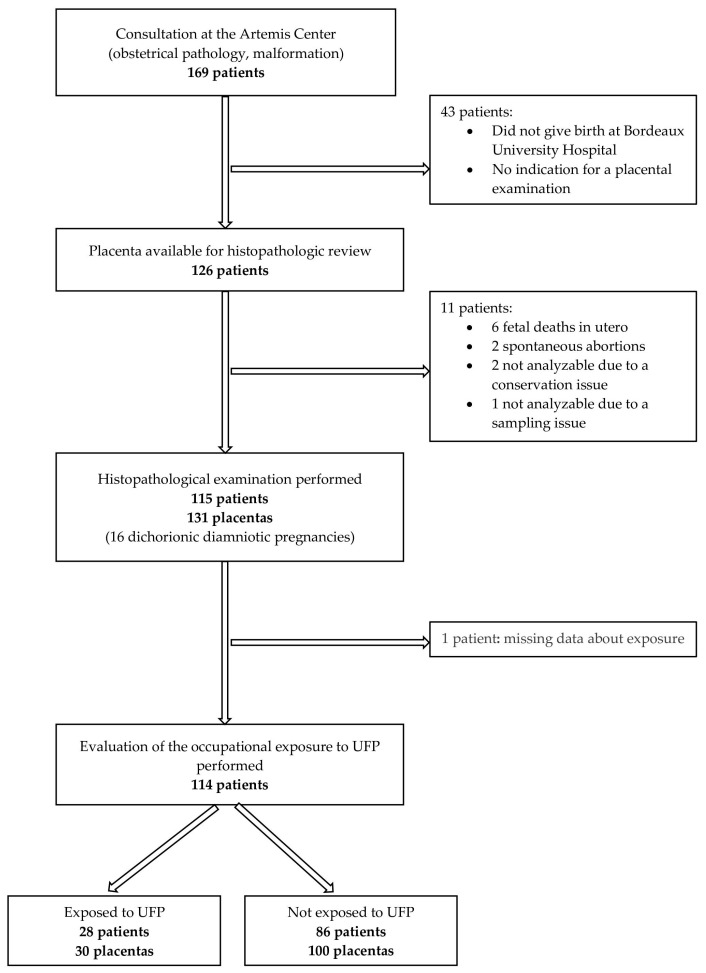
Flow chart.

**Table 1 ijerph-18-12719-t001:** Socio-demographic characteristics of the population.

Socio-Demographic Characteristics	Total		Unexposed		Exposed		*p*-Value *
Maternal age, median (*Q25*; *Q75*)	33	(28;36)	33	(28;36)	33	(25.8;35.2)	0.63
Maternal education, *n* (%)							
Short-cycle higher education	34	(30)	30	(35)	4	(14)	<0.01
Master’s degree or equivalent	30	(26)	28	(33)	2	(7.1)	
Professional qualification	22	(19)	13	(15)	2	(32)	
Professional high school	9	(7.9)	4	(4.7)	5	(18)	
High school	5	(4.4)	4	(4.7)	1	(3.6)	
Without certificate	6	(5.3)	3	(3.5)	3	(11)	
Unknown	8	(7)	14	(4.7)	4	(14)	
BMI category (kg/m^2^), *n* (%)							
<19	8	(7)	6	(7)	2	(7.1)	1
19–24.9	59	(52)	47	(55)	12	(43)	
25–29.9	17	(15)	13	(15)	4	(14)	
≥30	18	(17)	12	(14)	7	(25)	
Unknown	11	(9.6)	8	(9.3)	3	(11)	
Localization of home, *n* (%)							
Urban	70	(61)	54	(63)	16	(57)	1
Semi-urban	23	(20)	16	(19)	7	(25)	
Rural	18	(16)	13	(15)	5	(18)	
Unknown	3	(2.6)	3	(3.5)	0	(0)	
Chronic hypertension	4	(3.5)	4	(4.7)	0	(0)	1
Type of pregnancy, *n* (%)							
Singleton	93	(82)	67	(78)	26	(93)	0.85
Dichorionic diamniotic	16	(14)	14	(16)	2	(7.1)	
Monochorionic diamniotic	4	(3.5)	4	(4.7)	0	(0)	
Monochorionic monoamniotic	1	(0.8)	1	(1.2)	0	(0)	
Smoking during pregnancy, *n* (%)	20	(18)	12	(14)	8	(29)	0.18
Passive tobacco exposure ^1^, *n* (%)	10	(8.8)	8	(9.3)	2	(7.1)	0.62

^1^ Passive tobacco exposure: missing data in five cases (4.4%), all belong to the unexposed group. * Chi square or Fisher’s tests.

**Table 2 ijerph-18-12719-t002:** Obstetrical pathologies.

Obstetrical Pathologies	Total		Unexposed		Exposed		*p*-Value *
	*n*	(%)	*n*	(%)	*n*	(%)	
Assisted reproductive technology (ART)	13	(11)	11	(13)	2	(7.1)	1
Gestational diabetes	11	(9.6)	7	(8.1)	4	(14)	0.92
Gestational hypertension	4	(3.5)	3	(3.5)	1	(3.6)	1
Pre-eclampsia	37	(32)	28	(33)	9	(32)	0.97
Fetal growth restriction	50	(44)	39	(45)	11	(39)	0.57
Premature rupture of the membranes	25	(22)	19	(22)	6	(21)	0.94
Congenital malformation	21	(18)	14	(16)	7	(25)	0.3
Termination of pregnancy	12	(11)	7	(8.1)	5	(18)	0.33
Clinical suspicion of chorioamnionitis	10	(8.8)	8	(9.3)	2	(7.1)	1
Other pathology	22	(19)	16	(19)	6	(21)	0.74
Preterm birth							
37–32 WG	46	(40)	33	(38)	13	(46)	1
<32 WG	59	(52)	45	(52)	14	(50)	

* Chi square or Fisher’s tests.

**Table 3 ijerph-18-12719-t003:** Sources of the exposure to UFPs.

Occupation	Effective	
	*n*	(%)
Cleaner	10	35.7
Childcare professional	5	17.8
Agricultural worker	3	10.7
Cook	2	7
Laundry worker	1	3.6
Mail delivery	1	3.6
Maintenance engineer	1	3.6
Pharmaceutical mill worker	1	3.6
QHSE ^1^ worker in a paper mill	1	3.6
Chief of project in construction	1	3.6
Wood mill worker	1	3.6
Medical worker	1	3.6

^1^ QHSE: quality, health, security, environment.

**Table 4 ijerph-18-12719-t004:** Histological placental syndromes.

	Total		Unexposed		Exposed		*p*-Value
	*n*	(%)	*n*	(%)	*n*	(%)	
Maternal stromal-vascular malperfusion	120	(92)	89	(90)	31	(100)	0.23
Fetal stromal-vascular malperfusion	30	(23)	26	(26)	4	(13)	0.12
Ascending intra-uterine infection	22	(17)	20	(20)	2	(6.5)	0.08
Immune lesions	22	(17)	20	(20)	2	(6.5)	0.08

**Table 5 ijerph-18-12719-t005:** Detailed histological placental lesions.

	Total		Unexposed		Exposed		*p*-Value
	*n*	(%)	*n*	(%)	*n*	(%)	
Maternal stromal-vascular malperfusion							
Decidual arteriopathy	31	(24)	23	(23)	8	(26)	0.77
Accelerated villous maturation	99	(76)	67	(68)	23	(74)	0.49
Distal villous hypoplasia	17	(13)	14	(14)	3	(9.7)	0.76
Placental hypoplasia	53	(41)	34	(34)	19	(61)	<0.01
Villous infarction	30	(23)	23	(23)	7	(23)	0.94
Retroplacental hemorrhage	13	(10)	10	(10)	3	(10)	1
Fetal stromal-vascular malperfusion							
Thrombus in a major vessel	8	(6.2)	8	(8.1)	0	(0)	0.2
Villous stromal-vascular karyorrhexis	9	(7)	9	(9.1)	0	(0)	0.12
Avascular villosity	27	(21)	23	(23)	4	(13)	0.24
Ascending intra-uterine infection	22	(17)	20	(20)	2	(6.5)	0.08
Immune lesions							
Chronic intervillitis	9	(6.9)	9	(9.1)	0	(0)	0.11
Villitis of unknown etiology	12	(9.2)	12	(12)	0	(0)	0.07
Other immune lesion	9	(6.9)	7	(7.1)	2	(6.5)	1
Other lesions							
Crowding of villosity	105	(81)	81	(82)	24	(77)	0.59
Increased intervillous fibrin	104	(80)	80	(81)	24	(77)	0.68
Increased syncytial knots	104	(80)	79	(80)	25	(81)	0.92

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
