# Peer review of "Occupational Exposure to Ultrafine Particles and Placental Histopathological Lesions: A Retrospective Study about 130 Cases"

_ijerph, 2021, doi:10.3390/ijerph182312719_

Round 1

Reviewer 1 Report

I read with great interest the manuscript, which falls within the aim of this Journal. In my honest opinion, the topic is interesting enough to attract the readers’ attention. Nevertheless, authors should clarify some points and improve the discussion, as suggested below.

Authors should consider the following recommendations:

  • Manuscript should be further revised in order to correct some typos and improve style.
  • I suggest add further elements to highlight, at least briefly, how epigenetic changes (including the one potentially caused by ultrafine particles) may be one underlying cause for IUGR and hypertensive disorders of pregnancy (authors may refer to: PMID: 28466013; PMID: 28282763).

Author Response

Response to Reviewer 1 comments

Thank you for reviewing our manuscript. Here are the answers to your comments:

Point 1: Manuscript should be further revised in order to correct some typos and improve style.

Response 1: After further revisions, we corrected all the typos we could find.

Point 2: I suggest add further elements to highlight, at least briefly, how epigenetic changes (including the one potentially caused by ultrafine particles) may be one underlying cause for IUGR and hypertensive disorders of pregnancy (authors may refer to: PMID: 28466013; PMID: 28282763).

Response 2: We added some information about epigenetics and IUGR/hypertensive disorders (lines 305-309). Thank you for these interesting articles.

« Saenen and al. suggest that the mother’s exposition to air pollution and PM2.5 (particulate matter <2.5 µm, which includes UFP) are associated with some variations of the placental epigenetic mechanisms such as global DNA methylation or miRNA levels of expression [31]. Other studies suggest that some alterations in the expression of miRNA could be associated with higher risks of pre-eclampsia [32] or FGR [33]. »

Reviewer 2 Report

The overall quality of presentation  enhance the readability of the survey. While english is used generally correctly from a grammar and syntax point of view. 

The survey  fully clarify its position in the current literature in this domain 

The main contribution  fill the gaps that would open up challenging, interesting and significant research directions. 

The structure employed for the categorisation of this methods and techniques appears not to require further refinement.

There is a very nice attempt to discuss the advantages and limitations of the current techniques which are often presented in vague terms and with well structured references to the specifics.

Point for improve

The reference list, at the end, includes 29 references, but in the text there are 34,35....

see line 334(31)

see line 341(32,33)

see line 342 (34,35)

The authors have to write the rest of the references 

Author Response

Response to Reviewer 2 comments

Thank you for reviewing our manuscript. Here are the answers to your comments:

Point 1: The reference list, at the end, includes 29 references, but in the text there are 34,35....

see line 334(31)

see line 341(32,33)

see line 342 (34,35)

The authors have to write the rest of the references 

Response 1: we updated the reference list (1-37).

  1. Valentino, S.A.; Tarrade, A.; Aioun, J.; Mourier, E.; Richard, C.; Dahirel, M.; Rousseau-Ralliard, D.; Fournier, N.; Aubrière, M.-C.; Lallemand, M.-S.; et al. Maternal Exposure to Diluted Diesel Engine Exhaust Alters Placental Function and Induces Intergenerational Effects in Rabbits. Part Fibre Toxicol 2015, 13, 39, doi:10.1186/s12989-016-0151-7.
  2. Max Costa, Y.Y. Genetic and Epigenetic Effects of Nanoparticles. J Mol Genet Med 2013, 07, doi:10.4172/1747-0862.1000086.
  3. Xia, T.; Kovochich, M.; Brant, J.; Hotze, M.; Sempf, J.; Oberley, T.; Sioutas, C.; Yeh, J.I.; Wiesner, M.R.; Nel, A.E. Comparison of the Abilities of Ambient and Manufactured Nanoparticles To Induce Cellular Toxicity According to an Oxidative Stress Paradigm. Nano Lett. 2006, 6, 1794–1807, doi:10.1021/nl061025k.
  4. Saenen, N.D.; Martens, D.S.; Neven, K.Y.; Alfano, R.; Bové, H.; Janssen, B.G.; Roels, H.A.; Plusquin, M.; Vrijens, K.; Nawrot, T.S. Air Pollution-Induced Placental Alterations: An Interplay of Oxidative Stress, Epigenetics, and the Aging Phenotype? Clin Epigenet 2019, 11, 124, doi:10.1186/s13148-019-0688-z.
  5. Laganà, A.S.; Vitale, S.G.; Sapia, F.; Valenti, G.; Corrado, F.; Padula, F.; Rapisarda, A.M.C.; D’Anna, R. MiRNA Expression for Early Diagnosis of Preeclampsia Onset: Hope or Hype? The Journal of Maternal-Fetal & Neonatal Medicine 2018, 31, 817–821, doi:10.1080/14767058.2017.1296426.
  6. Chiofalo, B.; Laganà, A.S.; Vaiarelli, A.; La Rosa, V.L.; Rossetti, D.; Palmara, V.; Valenti, G.; Rapisarda, A.M.C.; Granese, R.; Sapia, F.; et al. Do MiRNAs Play a Role in Fetal Growth Restriction? A Fresh Look to a Busy Corner. BioMed Research International 2017, 2017, 1–8, doi:10.1155/2017/6073167.
  7. Burstyn, I.; Yang, Y.; Schnatter, A. Effects of Non-Differential Exposure Misclassification on False Conclusions in Hypothesis-Generating Studies. IJERPH 2014, 11, 10951–10966, doi:10.3390/ijerph111010951.
  8. Clemens, T.; Turner, S.; Dibben, C. Maternal Exposure to Ambient Air Pollution and Fetal Growth in North-East Scotland: A Population-Based Study Using Routine Ultrasound Scans. Environment International 2017, 107, 216–226, doi:10.1016/j.envint.2017.07.018.
  9. Lamichhane, D.K.; Ryu, J.; Leem, J.-H.; Ha, M.; Hong, Y.-C.; Park, H.; Kim, Y.; Jung, D.-Y.; Lee, J.-Y.; Kim, H.-C.; et al. Air Pollution Exposure during Pregnancy and Ultrasound and Birth Measures of Fetal Growth: A Prospective Cohort Study in Korea. Science of The Total Environment 2018, 619–620, 834–841, doi:10.1016/j.scitotenv.2017.11.058.
  10. Laurent, O.; Hu, J.; Li, L.; Kleeman, M.J.; Bartell, S.M.; Cockburn, M.; Escobedo, L.; Wu, J. Low Birth Weight and Air Pollution in California: Which Sources and Components Drive the Risk? Environment International 2016, 92–93, 471–477, doi:10.1016/j.envint.2016.04.034.
  11. Manangama, G.; Audignon-Durand, S.; Migault, L.; Gramond, C.; Zaros, C.; Teysseire, R.; Sentilhes, L.; Brochard, P.; Lacourt, A.; Delva, F. Maternal Occupational Exposure to Carbonaceous Nanoscale Particles and Small for Gestational Age and the Evolution of Head Circumference in the French Longitudinal Study of Children - Elfe Study. Environmental Research 2020, 185, 109394, doi:10.1016/j.envres.2020.109394.

Reviewer 3 Report

The manuscript is well written, it may require only a short editorial inspection regarding spelling and grammar. Some smaller wording related points have been detailed in the section of minor points.

Regarding methodology, implementation and statistics, there is not much to complain about from my side. It could be brought up that the way to estimate occupational exposure to UFP by simply categorizing professions with high and low risk levels is slightly to rough considering the low number of investigated cases. However, this fact does not disqualify the findings but only demands caution in judging the overall solidity of results.

When reading the manuscript, two times the presented numbers were confusing:

First, in line 241, the authors state that 94% of the children were born before 37 WG. The Table in line 252 then shows, that 40% of births happened 37-32 WG and 52% < 32 WG, that sums up to 92%. Why are 2% missing here?

The second confusion might be more severe in consequence: In Table 5, the authors show the relative counts of placental lesions. Placental hypoplasia (PH) is highlighted with for 53 cases in total. Villous infarction (VI) is shown as not significant for total 30 cases. However, if we sum the cases of exposed and unexposed, we get 45 cases and not 30, as provided in the table. Can it be that significance has simply not been reached for VI because of a wrongly low N number, as the percentual relation of exposed to unexposed appears to be more significant than for PH (34:61 vs 23:71). If this turns out to be the case, it would add another explanatory point for understanding the link between UFP and FGR, as it futher points towards vascular malperfusion as a possible cause of placental hypoplasia.

Minor points:

Line 128: "Work processes generating UFP..." is not really a good classification, as in many professions UFP are not generated de-novo, but only handled and whirled up by employees.

Line 343, wording: "It might also explain by a longer..."

Author Response

Response to Reviewer 3 comments

Thank you for reviewing our manuscript. Here are the answers to your comments:

Point 1: The manuscript is well written, it may require only a short editorial inspection regarding spelling and grammar. Some smaller wording related points have been detailed in the section of minor points.

Response 1: Thank you for your comment. After further revisions, we corrected all the typos we could find

Point 2: Regarding methodology, implementation and statistics, there is not much to complain about from my side. It could be brought up that the way to estimate occupational exposure to UFP by simply categorizing professions with high and low risk levels is slightly to rough considering the low number of investigated cases. However, this fact does not disqualify the findings but only demands caution in judging the overall solidity of results.

Response 2: Indeed, we agree with you. However, given the size of the sample, we have chosen to compare only two exposure groups and we have discussed the limitations that this may entail in the discussion (from line 349). This study is an exploratory study that needs to be confirmed by prospective cohort studies.

Point 3: When reading the manuscript, two times the presented numbers were confusing: First, in line 241, the authors state that 94% of the children were born before 37 WG. The Table in line 252 then shows, that 40% of births happened 37-32 WG and 52% < 32 WG, that sums up to 92%. Why are 2% missing here?

Response 3: Thank you for pointing this error. After checking the database:

  • 46 pregnancies were delivered between 32-37 WG (40.3%)
  • 59 pregnancies were delivered <32 WG (51.7%)
  • 9 pregnancies were delivered >37 WG (7.9%)
  • In total, 105 pregnancies were delivered <37 WG which represents 92.1% of the pregnancies. We corrected it in the text: line 253, 94% à 92%.

Point 4: The second confusion might be more severe in consequence: In Table 5, the authors show the relative counts of placental lesions. Placental hypoplasia (PH) is highlighted with for 53 cases in total. Villous infarction (VI) is shown as not significant for total 30 cases. However, if we sum the cases of exposed and unexposed, we get 45 cases and not 30, as provided in the table. Can it be that significance has simply not been reached for VI because of a wrongly low N number, as the percentual relation of exposed to unexposed appears to be more significant than for PH (34:61 vs 23:71). If this turns out to be the case, it would add another explanatory point for understanding the link between UFP and FGR, as it futher points towards vascular malperfusion as a possible cause of placental hypoplasia.

Response 4: After checking the database:

  • Villous infarctions were present in 30 placentas, 7 exposed and 23 not exposed.
  • The statistical analysis was made based on these numbers, so the p-value is not changed.
  • It was probably an error of copy and paste. We corrected it in the text:
    Table 5, 22 (71) à 7 (23)

Point 5: Minor points: Line 128: "Work processes generating UFP..." is not really a good classification, as in many professions UFP are not generated de-novo, but only handled and whirled up by employees.

Response 5: We changed the formulation to “work processes leading to an exposition to UFP” (line 128).

Point 6: Line 343, wording: "It might also explain by a longer..."

Response 6: We corrected the typo : “It might also be explained by a longer » (line 367)